# Design and Synthesis of D_3_R Bitopic Ligands with Flexible Secondary Binding Fragments: Radioligand Binding and Computational Chemistry Studies

**DOI:** 10.3390/molecules29010123

**Published:** 2023-12-24

**Authors:** Gui-Long Tian, Chia-Ju Hsieh, Michelle Taylor, Ji Youn Lee, Robert R. Luedtke, Robert H. Mach

**Affiliations:** 1Division of Nuclear Medicine and Clinical Molecular Imaging, Department of Radiology, Perelman School of Medicine, University of Pennsylvania, Philadelphia, PA 19104, USA; guilongtian6@gmail.com (G.-L.T.); chiahs@pennmedicine.upenn.edu (C.-J.H.);; 2Department of Pharmacology and Neuroscience, University of North Texas Health Science Center, Fort Worth, TX 76107, USA; michelle.taylor@unthsc.edu (M.T.);

**Keywords:** dopamine 2 receptor, dopamine 3 receptor, flexible linker, bitopic ligands, molecular dynamic simulation

## Abstract

A series of bitopic ligands based on Fallypride with a flexible secondary binding fragment (SBF) were prepared with the goal of preparing a D_3_R-selective compound. The effect of the flexible linker (**(*R***,***S*)*-trans*-2a**–**d**), SBFs (**(*R***,***S*)*-trans*-2h**–**j**), and the chirality of orthosteric binding fragments (OBFs) (**(*S***,***R*)*-trans*-d**, **(*S*,*R*)*-trans*-i**, **(*S***,***S*)*-trans*-d**, **(*S***,***S*)*-trans*-i, (*R***,***R*)*-trans*-d,** and **(*R***,***R*)*-trans*-i**) were evaluated in in vitro binding assays. Computational chemistry studies revealed that the interaction of the fragment binding to the SBF increased the distance between the pyrrolidine nitrogen and ASP110^3.32^ of the D_3_R, thereby reducing the D_3_R affinity to a suboptimal level.

## 1. Introduction

Dopamine receptors are a class of G protein-coupled receptors. Five subtypes of dopamine receptors have been identified, and they are divided into two classes based on their sequence and functional roles. The dopamine 1-like receptors include the dopamine 1 receptor (D_1_R) and the dopamine 5 receptor (D_5_R), and the dopamine 2-like receptors consist of the dopamine 2 receptor (D_2_R), the dopamine 3 receptor (D_3_R), and the dopamine 4 receptor (D_4_R) [1,2,3]. Previous studies showed that the dysregulation of D_2_R and D_3_R is related to many central nervous system (CNS) diseases [4,5]. These two dopamine receptors have been used as therapeutic targets for treating neurological and neuropsychiatric disorders, including schizophrenia, drug addiction, and Parkinson’s disease (PD) [4,6,7,8].

The D_2_R and D_3_R have a differential distribution in the human brain [9,10,11]. For example, in the globus pallidus internal part, thalamus, red nucleus, and substantia nigra, the D_3_R has a higher density than the D_2_R [12,13]. Moreover, an autoradiography study in chronic cocaine abuse showed that the density of the D_2_R and D_3_R changed differently (the D_2_R had no change vs. the D_3_R increased) [4,14,15,16,17,18]. An in vitro autoradiography study of PD brain samples yielded similar results as above [12,14]. These results indicate that these two dopamine receptors play different roles in the CNS; the D_3_R is thought to play a key role in mediating *L*-DOPA-induced dyskinesia [16].

Positron emission tomography (PET) is a functional imaging technique that uses radiotracers to image receptors in the CNS. Over the past several years, [^18^F]Fallypride [19,20,21,22], [^11^C]Raclopride [23,24,25,26,27], [^11^C]FLB 457 [20,26,28], and [^11^C]-(+)-PHNO [29,30,31] have been used as radiotracers for PET imaging studies of the D_2_-like receptors in humans (Figure 1). Unfortunately, all four radiotracers cannot image the D_3_R independently of the D_2_R. The lack of highly selective D_3_R PET radiotracers has prevented the imaging of the D_3_R independently of the D_2_R. Therefore, developing a D_3_R selective radiotracer that can bind independently of the D_2_R is of high importance to further understand the behavior of the D_3_R in the CNS.

The high structural similarity of the D_2_R and D_3_R (78% similar in the transmembrane spanning region) [4], has presented a challenge in the development of a D_3_R versus a D_2_R selective ligand [32]. During the past decade, a number of D_3_R selective ligands with the phenylpiperazine structure have been reported. Our group has reported Fluortriopride (FTP; LS-3-134) [33] (Figure 2), which has high D_3_R affinity (*K*_i_ = 0.17 nM) and good selectivity versus the D_2_R (163-fold). The 1,2,4-triazole-based scaffold also has notable D_3_R versus D_2_R selectivity. For example, GSK598,809 (D_3_R *K*_i_ = 3.2 nM) [34] represents a D_3_R selective ligand with this scaffold and has a 670-fold D_3_R vs. D_2_R selectivity. Based on the structure of tranylcypromine, Chen et al. [35] reported a D_3_R ligand (CJ-1882) with 2.8 nM affinity for the D_3_R and 223-fold D_3_R versus D_2_R selectivity. Recently, we reported a flexible scaffold structure (HY-2-93) with 0.8 nM affinity for the D_3_R and 180-fold D_3_R versus D_2_R selectivity [36].

This report describes the continuation of our effort to identify D_3_R antagonists having a high selectivity versus the D_2_R as a potential PET radiotracer for in vivo imaging studies. Our recent structure–activity relationship (SAR) studies based on Fallypride revealed some critical factors for designing a D_3_R versus D_2_R selective ligand [37]. These results, together with the flexible D_3_R ligands we reported previously [36], inspired us to design a D_3_R selective antagonist by introducing a flexible secondary binding fragment (SBF) based on Fallypride. 

## 2. Results

### 2.1. Chemistry 

As shown in Figure 1, we used a two-step route to prepare the Fallypride-based bitopic ligands. First, a mixture of **(*S*,*R*)*-trans*-1** and 1,1-carbonyldiimidazole (CDI) in acetonitrile was stirred overnight. Second, the corresponding amines reacted with the crude product from the previous step in toluene at 100 °C, delivering the designed bitopic ligands **(*R*,*S*)*-trans*-2a**–**j** in good yield.

Next, in Figure 2, the chirality of orthosteric binding fragments (OBFs) was explored. The diastereomers and enantiomers of **(*R*,*S*)*-trans*-2d** and **(*R*,*S*)*-trans*-2i** were prepared. **(*R*,*S*)*-trans*-1**, **(*S*,*S*)*-trans*-1** or **(*R*,*R*)*-trans*-1**, the diastereomers and enantiomers of **(*S*,*R*)*-trans*-1**, reacted with CDI in acetonitrile. The generated intermediate was further treated with the corresponding amines in toluene at 100 °C to give the diastereomers and enantiomers of **(*R*,*S*)*-trans*-2d** and **(*R*,*S*)*-trans*-2i**.

### 2.2. SAR Study of Flexible Bitopic Ligands towards D_2_R and D_3_R 

The compounds synthesized in Figure 1 and Figure 2 were submitted for in vitro binding assays measuring their affinity for D_3_R and D_2_R. The results of the binding assays are shown in Table 1 and Table 2.

Ligands with a flexible aliphatic linker (i.e., **(*R*,*S*)*-trans*-2a**–**d**) have a high affinity for the D_3_R; as the length of the aliphatic linkers increased, their affinity for the D_2_R decreased. These results suggest that bitopic ligands with longer aliphatic linkers increase the selectivity for the D_3_R versus the D_2_R. Bitopic ligands with a polyethylene glycol (PEG) as a flexible linker were also evaluated (**(*R*,*S*)*-trans*-2e**–**g**). Although these bitopic ligands maintained high affinity for the D_3_R, they did not have the expected higher D_3_R versus D_2_R selectivity. The ligands **(*R*,*S*)*-trans*-2h**–**j**, which have a flexible linker containing a thiophene in the SBF, were evaluated for their affinity for the D_2_R and D_3_R. The 2-thiophene analog with a 6-carbon spacer, **(*R*,*S*)*-trans*-2h**, had a sub-nanomolar affinity for the D_3_R (K_i_ = 0.6 ± 0.1 nM); increasing the spacer group to 8 carbon atoms did not affect the D_3_R affinity (K_i_ = 0.7 ± 0.1 nM). The analog having a 6-carbon spacer with a 3-thiophene had lower D_3_R affinity relative to its 2-thiophene congener, **(*R*,*S*)*-trans*-2h**.

We next evaluated the effect of the chirality of OBFs (Table 2). The enantiomers and diastereomers of **(*R***,***S*)*-trans*-2d** and **(*R***,***S*)*-trans*-2i** were utilized for this as these two ligands have comparatively better affinity for the D_3_R. Changing the chiral centers of OBFs has a dramatic effect on their affinity for the D_2_R and D_3_R. **(*S***,***R*)*-trans*-2d** is an enantiomer of **(*R***,***S*)*-trans*-2d**. The affinity for the D_2_R and the D_3_R decreased one thousand-fold to 1945 nM and 2628 nM, respectively. Meanwhile, the D_2_R and D_3_R affinity of **(*S***,***S*)*-cis*-2d** and **(*R***,***R*)*-cis*-2d**, the diastereomers of **(*R***,***S*)*-trans*-2d**, were decreased by one hundred to one thousand-fold. These results were also observed for the enantiomer and diastereomers of **(*R***,***S*)*-trans*-2i**.

### 2.3. β-Arrestin Competition Assay 

A β-arrestin competition assay was conducted to determine the potency of the above-mentioned compounds for competing with dopamine (30 nM) for the D_3_R. High potency in this assay (EC50 ~2 nM) is needed in order to compete with synaptic dopamine for binding to the D_3_R in vivo [39]. As shown in Table 1, **(*R***,***S*)*-trans*-2a** has high potency in the β-arrestin assay. Compounds having a longer aliphatic linker (**(*R***,***S*)*-trans*-2b**–**d**) have a similar IC_50_ value in this assay. As expected, the IC_50_ value of β-arrestin assay parallels the K_i_ value in the D_3_R binding assay. It is of interest to note that the potency of **(*R***,***S*)*-trans*-2a** in the β-arrestin assay is higher than its potency in the in vitro binding assay for the D_3_R.

### 2.4. Molecular Dynamic Simulation (MDS) Studies

The representative binding poses for the D_3_R or D_2_R from the MDS production runs for four compounds with different lengths of aliphatic groups in the SBFs (**(*R***,***S*)*-trans*-2a**, **(*R***,***S*)*-trans*-2b**, **(*R***,***S*)*-trans*-2c**, and **(*R***,***S*)*-trans*-2d**) are shown in Figure 3. The “Fallypride” fragment of all four compounds posed in the orthosteric binding pocket interacted with the amino acid residues at transmembrane (TM) 3, 5, and 6 in both the D_2_R and the D_3_R. The SBF of all four compounds interacted with the amino acid residues in TM 2 and 7 in the secondary binding site for the D_3_R. For the D_2_R, the SBF of the four compounds interacted with the secondary binding pocket (TM 2 and 7) or the loop region, EC1 or EC2. Hydrogen bonds formed between ASP110^3.32^ and the protonated nitrogen in the pyrrolidine ring, the key interaction that has been reported for the D_3_R [39,40], were observed in all four compounds in the D_3_R (Figure 3A–D). The benzene rings of **(*R***,***S*)*-trans*-2a**, **(*R***,***S*)*-trans*-2b**, and **(*R***,***S*)*-trans*-2c** formed a π-stacking interaction with PHE345^6.51^ in the orthosteric binding pocket of the D_3_R (Figure 3A–C). For the D_2_R, a hydrogen bond or salt bridge formed between ASP114^3.32^ and the protonated nitrogen in the pyrrolidine ring (the key interaction that has been reported for D_2_R [41,42]) was observed in the MDS for all four compounds (Figure 3F–I). The absence of a π-stacking interaction in the orthosteric binding site of the D_2_R may partially explain the higher affinity of the four compounds for the D_3_R. 

The distance between the protonated nitrogen in the pyrrolidine ring of six ligands (**(*R***,***S*)*-trans*-2a-d**, **(*R***,***S*)*-trans*-2h**, and **(*R***,***S*)*-trans*-2i**) and ASP110^3.32^ in the D_3_R was in the range of 3.28 to 3.79 Å (Table 3); this distance is greater than that with Fallypride (3.17 ± 0.21 Å) [39]. This indicates that these compounds have weaker interactions with ASP110^3.32^ in the D_3_R compared to Fallypride, resulting in a reduction in the binding affinity for the D_3_R. Furthermore, the distance between ASP114^3.32^ in the D_2_R and the protonated nitrogen in the pyrrolidine ring of the six ligands was greater (3.45 to 9.32 Å; Table 3) and had a higher standard deviation (0.94 to 2.20 Å; Table 3); these results are consistent with the low stability of these compounds in the binding pocket and are reflected in the lower binding affinity of the six compounds for the D_2_R.

Another method for studying ligand–protein interactions is to measure the frequency of the contact between the ligand and amino acid residues in the ligand binding site. The summary of the frequency of contacts for each compound in the D_3_R or D_2_R is shown in Figure 4. A stable salt bridge or hydrogen bound was formed between ASP110^3.32^ in the D_3_R and the pyrrolidine nitrogen in all six compounds (frequency of contact > 0.8; Figure 4A). Consequently, **(*R***,***S*)*-trans*-2a-d**, **(*R***,***S*)*-trans*-2h**, and **(*R***,***S*)*-trans*-2i** all had high affinity for the D_3_R (0.6–3.4 nM). For the D_2_R, **(*R***,***S*)*-trans*-2a-d**, **(*R***,***S*)*-trans*-2h**, and **(*R***,***S*)*-trans*-2i** displayed a poor to moderate probability of forming a hydrogen bound with ASP114^3.32^ (frequency of contact = 0.01–0.81; Figure 4B). There was also a wide range in the probability of forming hydrophobic interactions (0.15–0.87) in the orthosteric binding pocket (TM 3 and 5; Figure 4B). The low interaction with ASP114^3.32^ and variable hydrophobic interactions likely explain the high range in the D_2_R affinity for the six compounds.

Although there were differences in the interaction of **(*R***,***S*)*-trans*-2a-d**, **(*R***,***S*)*-trans*-2h**, and **(*R***,***S*)*-trans*-2i** with VAL86^2.61^ in the secondary binding pocket of the D_3_R, this interaction was inconsequential since all six compounds had similar affinity for the D_3_R. On the other hand, the interaction of the SBF with the D_2_R may play a role in the D_2_R affinity of the six ligands. **(*R***,***S*)*-trans*-2c**, **(*R***,***S*)*-trans*-2d**, **(*R***,***S*)*-trans*-2h**, and **(*R***,***S*)*-trans*-2i**, which have a longer length of the SBF (8 and 10 carbon spacer group), displayed a higher frequency of contact of TRP100^EC1^ in the loop between TM 2 and 3 (EC1) and ILE184^EC2^ in the loop between TM 4 and 5 (EC2; Figure 4B). The favorable interaction of **(*R***,***S*)*-trans*-2c** and **(*R***,***S*)*-trans*-2d** with the loop region of EC2 interferes with the formation of the hydrogen bond with ASP114^3.32^, and this likely explains the lower D_2_R affinity of **(*R***,***S*)*-trans*-2c** and **(*R***,***S*)*-trans*-2d**.

## 3. Discussion

We prepared a panel of bitopic D_3_R ligands based on Fallypride with the goal of developing a D_3_R selective antagonist. The most D_3_R selective compounds were **(*R*,*S*)*-trans*-2c** and **(*R*,*S*)*-trans*-2d**, which had a ~3 nM affinity for the D_3_R and 5–10-fold selectivity versus the D_2_R. This selectivity was attributed to their reduced affinity for the D_2_R, which was likely caused by the lower interactions in the orthosteric binding site of the D_2_R. **(*R*,*S*)*-trans*-2a** had higher potency (IC_50_ = 0.8 ± 0.5 nM) to compete with dopamine than Fallypride (IC_50_ = 1.7 ± 0.8 nM) in the β-arrestin recruitment assay, which was unexpected given its lower affinity for the D_3_R relative to Fallypride. The reason for this discrepancy between the receptor affinity and potency in the β-arrestin competition assay is not clear. **(*R*,*S*)*-trans*-2d** had modest D_3_R versus D_2_R selectivity (~12.7-fold) and a much better ability to compete with endogenous dopamine (IC_50_ = 21.2 ± 9.8 nM) than the highly D_3_R-selective radiotracer FTP (IC_50_ = 611.7 ± 101.3 nM). Increasing the length and steric bulk of the flexible linker in the SBF did not improve the D_3_R versus D_2_R selectivity. This structural modification also reduced the affinity of the bitopic compounds for both the D_3_R and the D_2_R relative to that of Fallypride, which is largely due to the effect of the substituent in the SBF increasing the distance between the pyrrolidine nitrogen and the key ASP^3.32^ residues in the orthosteric binding site of the D_3_R and the D_2_R. Consequently, the SAR studies described above indicate that the modification of Fallypride to contain an SBF does not improve the selectivity of this scaffold that is needed to generate a D_3_R-selective PET radiotracer. 

## 4. Materials and Methods

### 4.1. Chemistry

The starting materials and anhydrous solvents were purchased from Sigma-Aldrich, TCI America, Alfa Aesar, and Ambeed and were used without further purification. 5-(3-fluoropropyl)-2,3-dimethoxybenzoic acid was prepared using a reported method. The NMR spectra were taken on a Bruker DMX 400 MHz. The chemical shifts (δ) in the NMR spectra (^1^H and ^13^C) were referenced by assigning the residual solvent peaks. The purification of organic compounds was carried out on a Biotage Isolera One with a dual-wavelength UV−vis detector (silica gel: 230−400 mesh, 60 Å). The compound structures and identity were confirmed by ^1^H- and ^13^C NMR and mass spectrometry (Appendix A). The procedure for the synthesis of 1 can be found in our previous report [37].

#### General Methods for the Synthesis of **2**

A mixture of 1 (0.2 mmol) and 1,1′-Carbonyldiimidazole (CDI) (0.2 mmol) in acetonitrile (4 mL) was stirred at room temperature overnight. The solvent was removed under a vacuum, and the crude product was used for the next step directly.

A solution of the crude product mentioned above and the corresponding amine (0.4 mmol) in toluene was heated to 100 °C overnight. The solvent was removed under vacuum, and the crude product was purified by flash silica chromatography (CH_2_Cl_2_/MeOH = 95:5) to afford **2**.

*(3R,5S)-1-allyl-5-((5-(3-fluoropropyl)-2,3-dimethoxybenzamido)methyl)pyrrolidin-3-yl butylcarbamate* (**(*R*,*S*)-*trans*-2a**). A total of 49 mg colorless oil, 51% yield. ^1^H NMR (400 MHz, CDCl_3_) δ 8.40 (s, 1H), 7.53 (d, *J* = 2.1 Hz, 1H), 6.87 (d, *J* = 2.1 Hz, 1H), 6.00–5.72 (m, 1H), 5.21 (d, *J* = 17.1 Hz, 1H), 5.12 (d, *J* = 9.4 Hz, 1H), 5.05 (s, 1H), 4.62 (s, 1H), 4.51 (t, *J* = 5.9 Hz, 1H), 4.39 (t, *J* = 5.9 Hz, 1H), 3.89 (s, 3H), 3.87 (s, 3H), 3.85–3.81 (m, 1H), 3.54–3.48 (m, 2H), 3.36 (s, 1H), 3.14 (q, *J* = 6.7 Hz, 2H), 2.98 (s, 2H), 2.77–2.66 (m, 2H), 2.42 (s, 1H), 2.08–1.90 (m, 4H), 1.50–1.41 (m, 2H), 1.39–1.28 (m, 2H), 0.91 (t, *J* = 7.3 Hz, 3H); ^13^C NMR (101 MHz, CDCl_3_) δ 165.7, 156.1, 152.6, 146.0, 137.5, 135.2, 126.4, 122.3, 117.6, 115.8, 83.2 (d, *J* = 164.9 Hz), 72.7, 61.5, 60.6, 59.7, 56.5, 56.2, 40.8, 39.8, 35.8, 32.2, 32.0 (d, *J* = 19.8 Hz), 31.3 (d, *J* = 5.4 Hz), 20.0, 13.8. HRMS (ESI) calculated for C_25_H_38_FN_3_O_5_Na^+^ ([M + Na^+^]) 502.2693, found: 502.2688.

*(3R,5S)-1-allyl-5-((5-(3-fluoropropyl)-2,3-dimethoxybenzamido)methyl)pyrrolidin-3-yl hexylcarbamate* (**(*R*,*S*)-*trans*-2b**). A total of 42 mg colorless oil, 41% yield. ^1^H NMR (400 MHz, CDCl_3_) δ 8.42 (s, 1H), 7.53 (d, *J* = 2.1 Hz, 1H), 6.87 (d, *J* = 2.1 Hz, 1H), 5.89 (s, 1H), 5.22 (d, *J* = 17.2 Hz, 1H), 5.13 (d, *J* = 9.1 Hz, 1H), 5.06 (s, 1H), 4.62 (s, 1H), 4.51 (t, *J* = 5.9 Hz, 1H), 4.39 (t, *J* = 5.9 Hz, 1H), 3.89 (s, 7H), 3.60–3.50 (t, *J* = 6.5 Hz, 2H), 3.37 (s, 1H), 3.14 (q, *J* = 6.8 Hz, 2H), 2.98 (s, 2H), 2.77–2.70 (m, 2H), 2.43 (s, 1H), 2.09–1.91 (m, 4H), 1.50–1.43 (m, 2H), 1.32–1.25 (m, 6H), 0.88 (t, *J* = 6.8 Hz, 3H); ^13^C NMR (101 MHz, CDCl_3_) δ 165.8, 156.0, 152.6, 146.1, 137.5, 135.3, 126.3, 122.3, 117.7, 115.9, 83.2 (d, *J* = 164.9 Hz), 72.7, 61.5, 60.5, 59.7, 56.2, 41.1, 39.8, 35.8, 32.0 (d, *J* = 19.8 Hz), 31.6, 31.3 (d, *J* = 5.3 Hz), 30.0, 26.5, 22.7, 14.1. HRMS (ESI) calculated for C_27_H_43_FN_3_O_5_ ^+^ ([M + H^+^]) 508.3181, found: 508.3178.

*(3R,5S)-1-allyl-5-((5-(3-fluoropropyl)-2,3-dimethoxybenzamido)methyl)pyrrolidin-3-yl octylcarbamate* (**(*R*,*S*)-*trans*-2c**). A total of 60 mg colorless oil, 56% yield. ^1^H NMR (400 MHz, CDCl_3_) δ 8.41 (s, 1H), 7.53 (d, *J* = 2.2 Hz, 1H), 6.87 (d, *J* = 2.2 Hz, 1H), 5.96–5.81 (m, 1H), 5.22 (d, *J* = 17.1 Hz, 1H), 5.13 (d, *J* = 9.3 Hz, 1H), 5.06 (s, 1H), 4.63 (s, 1H), 4.51 (t, *J* = 5.9 Hz, 1H), 4.39 (t, *J* = 5.9 Hz, 1H), 3.88 (s, 7H), 3.63–3.47 (m, 2H), 3.37 (s, 1H), 3.13 (q, *J* = 6.8 Hz, 2H), 2.98 (s, 2H), 2.76–2.70 (m, 2H), 2.43 (s, 1H), 2.09–1.88 (m, 4H), 1.50–1.43 (m, 2H), 1.31–1.25 (m, 10H), 0.89–0.84 (m, 3H); ^13^C NMR (101 MHz, CDCl_3_) δ 165.8, 156.1, 152.6, 146.1, 137.5, 135.1, 126.3, 122.3, 117.6, 115.8, 83.2 (d, *J* = 164.9 Hz), 72.6, 61.5, 60.6, 59.7, 56.2, 41.1, 39.8, 35.8, 32.0 (d, *J* = 19.8 Hz), 31.9, 31.3 (d, *J* = 5.3 Hz), 29.3, 29.3, 26.9, 22.8, 14.2. HRMS (ESI) calculated for C_29_H_47_FN_3_O_5_^+^ ([M + H^+^]) 536.3494, found: 536.3506.

*(3R,5S)-1-allyl-5-((5-(3-fluoropropyl)-2,3-dimethoxybenzamido)methyl)pyrrolidin-3-yl decylcarbamate* (**(*R*,*S*)-*trans*-2d**) and *(3S,5R)-1-allyl-5-((5-(3-fluoropropyl)-2,3-dimethoxybenzamido)methyl)pyrrolidin-3-yl decylcarbamate* (**(*S*,*R*)-*trans*-2d**). A total of 56 mg colorless oil, 50% yield. ^1^H-NMR (400 MHz, CDCl_3_) δ 8.47–8.32 (m, 1H), 7.53 (d, *J* = 2.2 Hz, 1H), 6.86 (d, *J* = 2.1 Hz, 1H), 5.94–5.80 (m, 1H), 5.20 (d, *J* = 17.1 Hz, 1H), 5.12 (d, *J* = 10.0 Hz, 1H), 5.05 (s, 1H), 4.64 (s, 1H), 4.50 (t, *J* = 5.9 Hz, 1H), 4.38 (t, *J* = 5.9 Hz, 1H), 3.88–3.80 (m, 7H), 3.59–3.45 (m, 2H), 3.36 (s, 1H), 3.13 (q, *J* = 6.8 Hz, 2H), 2.98 (s, 2H), 2.75–2.71 (m, 2H), 2.41 (s, 1H), 2.09–1.88 (m, 4H), 1.50–1.43 (m, 2H), 1.30–1.24 (m, 14H), 0.86 (t, *J* = 6.7 Hz, 3H); ^13^C NMR (101 MHz, CDCl_3_) δ 165.7, 156.1, 152.6, 146.0, 137.5, 135.2, 126.3, 122.3, 117.6, 115.8, 83.1 (d, *J* = 165.0 Hz), 72.6, 61.4, 60.6, 59.7, 56.5, 56.2, 41.1, 39.8, 35.8, 32.0, 32.0 (d, *J* = 19.8 Hz), 31.3 (d, *J* = 5.2 Hz), 30.1, 29.6, 29.4, 29.4, 26.9, 22.8, 14.2. HRMS (ESI) calculated for C_31_H_51_FN_3_O_5_^+^ ([M + H^+^]) 564.3807, found: 564.3838.

*(3R,5S)-1-allyl-5-((5-(3-fluoropropyl)-2,3-dimethoxybenzamido)methyl)pyrrolidin-3-yl (2-(2-(2-methoxyethoxy)ethoxy)ethyl)carbamate* (**(*R*,*S*)-*trans*-2e**). A total of 46 mg colorless oil, 40% yield. ^1^H NMR (400 MHz, CDCl_3_) δ 8.40 (s, 1H), 7.51 (d, *J* = 2.1 Hz, 1H), 6.85 (d, *J* = 2.2 Hz, 1H), 5.91–5.85 (m, 1H), 5.27 (s, 1H), 5.20 (d, *J* = 17.1 Hz, 1H), 5.11 (d, *J* = 10.2 Hz, 1H), 5.04 (s, 1H), 4.49 (t, *J* = 5.9 Hz, 1H), 4.37 (t, *J* = 5.9 Hz, 1H), 3.87–3.79 (d, *J* = 3.2 Hz, 7H), 3.70–3.59 (m, 9H), 3.54–3.50 (m, 5H), 3.36 (s, 3H), 3.32 (q, *J* = 5.2 Hz, 2H), 2.98 (s, 2H), 2.71 (t, *J* = 7.7 Hz, 2H), 2.43 (s, 1H), 2.07–1.91 (m, 4H); ^13^C NMR (101 MHz, CDCl_3_) δ 165.7, 156.2, 152.6, 146.0, 137.4, 135.0, 126.3, 122.2, 117.8, 115.8, 83.1 (d, *J* = 164.9 Hz), 72.7, 72.0, 70.5_9_, 70.5_6_, 70.3, 70.1, 61.4, 60.6, 59.6, 59.1, 56.5, 56.2, 40.8, 39.7, 35.7, 31.9 (d, *J* = 19.8 Hz), 31.3 (d, *J* = 5.2 Hz). HRMS (ESI) calculated for C_28_H_45_FN_3_O_8_^+^ ([M + H^+^]) 570.3185, found: 570.3204.

*(3R,5S)-1-allyl-5-((5-(3-fluoropropyl)-2,3-dimethoxybenzamido)methyl)pyrrolidin-3-yl (2,5,8,11-tetraoxatridecan-13-yl)carbamate* (**(*R*,*S*)-*trans*-2f**). A total of 48 mg colorless oil, 39% yield. ^1^H NMR (400 MHz, CDCl_3_) δ 8.42 (s, 1H), 7.52 (s, 1H), 6.87 (d, *J* = 2.2 Hz, 1H), 5.89 (s, 1H), 5.36–4.97 (m, 4H), 4.51 (t, *J* = 5.9 Hz, 1H), 4.39 (t, *J* = 5.9 Hz, 1H), 3.89 (s, 7H), 3.72–3.57 (m, 12H), 3.55–3.53(m, 5H), 3.37 (s, 3H), 3.36–3.30 (m, 2H), 3.10–2.86 (m, 2H), 2.76–2.71 (m, 2H), 2.43 (s, 1H), 2.12–1.88 (m, 4H); ^13^C NMR (101 MHz, CDCl_3_) δ 165.7, 156.1, 152.5, 146.0, 137.4, 134.9, 126.2, 122.2, 117.8, 115.7, 83.1 (d, *J* = 164.9 Hz), 72.6, 72.0, 70.6_3_, 70.6_1_, 70.5_8_, 70.5_1_, 70.3, 70.1, 61.4, 60.6, 59.6, 59.1, 56.5, 56.1, 40.8, 39.7, 35.7, 31.9 (d, *J* = 19.7 Hz), 31.2 (d, *J* = 5.2 Hz). HRMS (ESI) calculated for C_30_H_48_FN_3_O_9_Na^+^ ([M + H^+^]) 636.3267, found: 636.3267.

*(3R,5S)-1-allyl-5-((5-(3-fluoropropyl)-2,3-dimethoxybenzamido)methyl)pyrrolidin-3-yl (2-(2-(prop-2-yn-1-yloxy)ethoxy)ethyl)carbamate* (**(*R*,*S*)-*trans*-2g**). A total of 47 mg colorless oil, 43% yield. ^1^H NMR (400 MHz, CDCl_3_) δ 8.41 (s, 1H), 7.52 (d, *J* = 2.1 Hz, 1H), 6.86 (d, *J* = 2.1 Hz, 1H), 5.97–5.80 (m, 1H), 5.26–5.02 (m, 4H), 4.50 (t, *J* = 5.8 Hz, 1H), 4.39 (t, *J* = 5.8 Hz, 1H), 4.19 (s, 2H), 3.88– 3.81 (m, 7H), 3.71–3.62 (m, 5H), 3.59–3.48 (m, 4H), 3.35 (q, *J* = 5.5 Hz, 2H), 3.00 (s, 2H), 2.73 (t, *J* = 7.7 Hz, 2H), 2.45–2.44 (m, 2H), 2.11–1.91 (m, 4H); ^13^C NMR (101 MHz, CDCl_3_) δ 165.7, 156.1, 152.6, 146.0, 137.5, 135.2, 126.3, 122.3, 117.7, 115.8, 83.2 (d, *J* = 164.9 Hz), 79.6, 74.9, 72.8, 70.3, 70.2, 69.2, 61.5, 60.6, 59.6, 58.6, 56.6, 56.2, 40.8, 39.8, 35.8, 32.0 (d, *J* = 19.7 Hz), 31.3 (d, *J* = 5.2 Hz). HRMS (ESI) calculated for C_28_H_41_FN_3_O_7_^+^ ([M + H^+^]) 550.2923, found: 550.2917.

*(3R,5S)-1-allyl-5-((5-(3-fluoropropyl)-2,3-dimethoxybenzamido)methyl)pyrrolidin-3-yl (6-(thiophen-2-yl)hexyl)carbamate* ((***R*,*S*)-*trans*-2h**). A total of 73 mg colorless oil, 62% yield. ^1^H-NMR (400 MHz, CDCl_3_) δ 8.40 (d, *J* = 5.9 Hz, 1H), 7.53 (d, *J* = 2.2 Hz, 1H), 7.09 (dd, *J* = 5.2, 1.2 Hz, 1H), 6.91–6.85 (m, 2H), 6.76–6.74 (m, 1H), 5.91–5.81 (m, 1H), 5.20 (dd, *J* = 17.0, 1.8 Hz, 1H), 5.11 (d, *J* = 10.2 Hz, 1H), 5.04 (s, 1H), 4.67 (s, 1H), 4.50 (t, *J* = 5.9 Hz, 1H), 4.38 (t, *J* = 5.9 Hz, 1H), 3.89–3.81 (m, 7H), 3.58–3.44 (m, 2H), 3.38–3.27 (m, 1H), 3.13 (q, *J* = 6.7 Hz, 2H), 3.04–2.90 (m, 2H), 2.80 (t, *J* = 7.6 Hz, 2H), 2.76–2.69 (m, 2H), 2.41 (s, 1H), 2.09–1.91 (m, 4H), 1.70–1.63 (m, 2H), 1.51–1.44 (m, 2H), 1.38–1.32 (m, 4H); ^13^C NMR (101 MHz, CDCl_3_) δ 165.7, 156.1, 152.6, 146.0, 145.6, 137.5, 135.2, 126.7, 126.3, 124.1, 122.9, 122.3, 117.6, 115.8, 83.1 (d, *J* = 165.0 Hz), 72.6, 61.4, 60.5, 59.7, 56.4, 56.2, 41.0, 39.7, 35.7, 32.0 (d, *J* = 19.7 Hz), 31.7, 31.3 (d, *J* = 5.2 Hz), 30.0, 29.9, 28.8, 26.5. HRMS (ESI) calculated forC_31_H_45_FN_3_O_5_S^+^ ([M + H^+^]) 590.3058, found: 590.3052.

*(3R,5S)-1-allyl-5-((5-(3-fluoropropyl)-2,3-dimethoxybenzamido)methyl)pyrrolidin-3-yl (8-(thiophen-2-yl)octyl)carbamate* (**(*R*,*S*)-*trans*-2i**) or *(3S,5R)-1-allyl-5-((5-(3-fluoropropyl)-2,3-dimethoxybenzamido)methyl)pyrrolidin-3-yl (8-(thiophen-2-yl)octyl)carbamate* (**(*S*,*R*)-*trans*-2i**). A total of 84 mg colorless oil, 68% yield. ^1^H NMR (400 MHz, CDCl_3_) δ 8.40 (d, *J* = 5.3 Hz, 1H), 7.54 (d, *J* = 2.1 Hz, 1H), 7.09 (d, *J* = 5.1 Hz, 1H), 6.92–6.89 (m, 1H), 6.87 (d, *J* = 2.1 Hz, 1H), 6.76 (d, *J* = 3.4 Hz, 1H), 5.94–5.80 (m, 1H), 5.21 (d, *J* = 17.1 Hz, 1H), 5.11 (d, *J* = 10.2 Hz, 1H), 5.05 (s, 1H), 4.64 (s, 1H), 4.51 (t, *J* = 5.9 Hz, 1H), 4.39 (t, *J* = 5.9 Hz, 1H), 3.88 (d, 6H), 3.85–3.79 (m, 1H), 3.57–3.47 (m, 2H), 3.40–3.27 (m, 1H), 3.13 (q, *J* = 6.8 Hz, 2H), 3.03–2.91 (m, 2H), 2.80 (t, *J* = 7.6 Hz, 2H), 2.77–2.66 (m, 2H), 2.41 (s, 1H), 2.11–1.87 (m, 4H), 1.69–1.62 (m, 2H), 1.50–1.45 (m, 2H), 1.35–1.29 (m, 8H); ^13^C NMR (101 MHz, CDCl_3_) δ 165.7, 156.1, 152.6, 146.0, 145.8, 137.5, 135.2, 126.7, 126.4, 124.0, 122.8, 122.3, 117.5, 115.8, 83.1 (d, *J* = 165.0 Hz), 72.7, 61.4, 60.5, 59.7, 56.5, 56.2, 41.1, 39.7, 35.8, 32.0 (d, *J* = 19.9 Hz), 31.8, 31.3 (d, *J* = 5.3 Hz), 30.1, 30.0, 29.3, 29.3, 29.1, 26.8. HRMS (ESI) calculated for C_33_H_48_FN_3_O_5_SNa^+^ ([M + Na^+^]) 618.3371, found: 618.3379.

*(3R,5S)-1-allyl-5-((5-(3-fluoropropyl)-2,3-dimethoxybenzamido)methyl)pyrrolidin-3-yl (8-(thiophen-3-yl)octyl)carbamate* (**(*R*,*S*)-*trans*-2j**). A total of 81 mg colorless oil, 66% yield. ^1^H-NMR (400 MHz, CDCl_3_) δ 8.42 (s, 1H), 7.53 (s, 1H), 7.23 (t, *J* = 4.0 Hz, 1H), 6.93–6.87 (m, 2H), 6.87 (s, 1H), 6.00–5.80 (m, 1H), 5.30–5.05 (m, 3H), 4.62 (s, 1H), 4.51 (t, *J* = 5.9 Hz, 1H), 4.39 (t, *J* = 5.9 Hz, 1H), 3.89 (s, 7H), 3.64–3.43 (m, 2H), 3.35 (s, 1H), 3.14 (q, *J* = 6.8 Hz, 2H), 3.03–2.88 (m, 2H), 2.73 (t, *J* = 7.7 Hz, 2H), 2.61 (t, *J* = 7.7 Hz, 2H), 2.43 (s, 1H), 2.08–1.92 (m, 4H), 1.62–1.57 (m, 2H), 1.49–1.43 (m, 2H), 1.33–1.30 (m, 8H); ^13^C NMR (101 MHz, CDCl_3_) δ 165.7, 155.9, 152.5, 146.0, 143.1, 137.4, 135.2, 128.3, 126.2, 125.1, 122.1, 119.8, 117.7, 115.8, 83.0 (d, *J* = 165.0 Hz), 72.5, 61.4, 59.5, 56.1, 41.0, 39.8, 35.7, 31.9 (d, *J* = 19.8 Hz), 31.2 (d, *J* = 5.3 Hz), 30.5, 30.2, 29.9, 29.3, 29.2_1_, 29.1_9_, 26.7. HRMS (ESI) calculated for C_33_H_48_FN_3_O_5_SNa^+^ ([M + Na^+^]) 640.3191, found: 640.3209.

*(3S,5S)-1-allyl-5-((5-(3-fluoropropyl)-2,3-dimethoxybenzamido)methyl)pyrrolidin-3-yl decylcarbamate* (**(*S*,*S*)-*cis*-2d**) and *(3R,5R)-1-allyl-5-((5-(3-fluoropropyl)-2,3-dimethoxybenzamido)methyl)pyrrolidin-3-yl decylcarbamate* (**(*R*,*R*)-*cis*-2d**). A total of 54 mg colorless oil, 48% yield. ^1^H NMR (400 MHz, CDCl_3_) δ 8.49 (s, 1H), 7.56 (s, 1H), 6.86 (s, 1H), 5.88–5.85 (m, 1H), 5.33 (s, 1H), 5.21 (d, *J* = 17.1 Hz, 1H), 5.13–5.07 (m, 2H), 4.52 (t, *J* = 5.9 Hz, 1H), 4.40 (t, *J* = 5.9 Hz, 1H), 3.87 (s, 6H), 3.68 (s, 1H), 3.49 (d, *J* = 14.0 Hz, 2H), 3.20 (d, *J* = 8.8 Hz, 1H), 3.10–2.81 (m, 4H), 2.74 (t, *J* = 7.7 Hz, 2H), 2.48–2.31 (m, 2H), 2.08 –1.95 (m, 2H), 1.77–1.73 (m, 2H), 1.30–1.24 (m, 16H), 0.87 (t, *J* = 6.6 Hz, 3H); ^13^C NMR (101 MHz, CDCl_3_) δ 165.7, 156.3, 152.6, 137.8, 135.2, 123.4, 122.4, 117.5, 115.3, 83.2 (d, *J* = 165.0 Hz), 72.3, 61.4, 60.1, 56.5, 55.9, 41.2, 41.0, 35.6, 32.0 (d, *J* = 19.9 Hz), 32.0, 31.4 (d, *J* = 5.3 Hz), 30.0, 29.7, 29.7, 29.5, 29.4, 26.9, 22.8, 14.2. HRMS (ESI) calculated for C_31_H_51_FN_3_O_5_^+^ ([M + H^+^]) 564.3807, found: 564.3806.

*(3S,5S)-1-allyl-5-((5-(3-fluoropropyl)-2,3-dimethoxybenzamido)methyl)pyrrolidin-3-yl (8-(thiophen-2-yl)octyl)carbamate* (**(*S*,*S*)-*cis*-2i**) and *(3R,5R)-1-allyl-5-((5-(3-fluoropropyl)-2,3-dimethoxybenzamido)methyl)pyrrolidin-3-yl (8-(thiophen-2-yl)octyl)carbamate* (**(*R*,*R*)-*cis*-2i**). A total of 69 mg colorless oil, 56% yield. ^1^H NMR (400 MHz, CDCl_3_) δ 8.49 (s, 1H), 7.57 (s, 1H), 7.10 (d, *J* = 5.0 Hz, 1H), 6.92–6.90 (m, 1H), 6.86 (s, 1H), 6.77 (d, *J* = 3.4 Hz, 1H), 5.86 (s, 1H), 5.33 (s, 1H), 5.21 (d, *J* = 17.1 Hz, 1H), 5.16–5.06 (m, 2H), 4.51 (t, *J* = 5.9 Hz, 1H), 4.40 (t, *J* = 5.9 Hz, 1H), 3.87 (s, 6H), 3.68 (s, 1H), 3.49 (s, 2H), 3.20 (s, 1H), 3.10–2.98 (m, 2H), 2.94–2.88 (m, 2H), 2.80 (t, *J* = 7.6 Hz, 2H), 2.74 (t, *J* = 7.8 Hz, 2H), 2.52–2.30 (m, 2H), 2.08–1.95 (m, 2H), 1.76 (d, *J* = 14.2 Hz, 1H), 1.77–1.62 (m, 4H), 1.38–1.23 (m, 9H); ^13^C NMR (101 MHz, CDCl_3_) δ 165.7, 156.4, 152.6, 145.9, 137.8, 135.5, 126.8, 124.1, 122.9, 122.4, 117.6, 115.3, 83.2 (d, *J* = 165.1 Hz), 72.4, 61.4, 60.1, 56.4, 56.0, 41.2, 41.0, 35.6, 32.0 (d, *J* = 19.8 Hz), 31.9, 31.4 (d, *J* = 5.2 Hz), 30.0, 30.0, 29.4, 29.4, 29.2, 26.9. HRMS (ESI) calculated for C_33_H_48_FN_3_O_5_SNa^+^ ([M + H^+^]) 640.3191, found: 640.3200.

### 4.2. Receptor Binding and β-Arrestin Assays

The receptor binding assay for the D_2_R and the D_3_R and the β-arrestin recruitment assay for the D_3_R were performed by following the previously reported methods [37]. 

### 4.3. Molecular Docking and Molecular Dynamics Simulation (MDS) Studies

The X-ray structures of the D_2_R (PDB ID: 6CM4, resolution 2.87 Å) and the D_3_R (PDB ID: 3PBL, resolution 2.89 Å) were obtained from the RCSB Protein Data Bank (https://www.rcsb.org/ (accessed on 12 November 2023)) to conduct the docking and MDS studies with **(*R*,*S*)*-trans*-2a**, **(*R*,*S*)*-trans*-2b**, **(*R*,*S*)*-trans*-2c**, **(*R*,*S*)*-trans*-2d**, **(*R*,*S*)*-trans*-2h**, and **(*R*,*S*)*-trans*-2i**. The docking and MDS studies were performed using the previously reported methods [37,39]. 

## 5. Conclusions

A panel of bitopic D_3_R ligands based on Fallypride was synthesized with the goal of developing a D_3_R selective ligand. The most selective compound, **(*R*,*S*)*-trans*-2d**, had a modest D_3_R versus D_2_R selectivity (~12.7-fold) and a much better ability to compete with endogenous dopamine (IC_50_ = 21.2 ± 9.8 nM) than the highly D_3_R-selective radiotracer FTP (IC_50_ = 611.7 ± 101.3 nM). Increasing the length and steric bulk of the flexible linker in the SBF did not improve the D_3_R versus D_2_R selectivity and resulted in a reduction in the affinity for the D_3_R. Computational chemistry studies revealed that this reduction in affinity was caused by an increase in the distance between the pyrrolidine nitrogen and ASP110^3.32^ in the D_3_R caused by the interaction of the substituents with the SBF.

## Data Availability

The article contains complete data used to support the findings of this study.

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
