# Peer review of "Design and Synthesis of D3R Bitopic Ligands with Flexible Secondary Binding Fragments: Radioligand Binding and Computational Chemistry Studies"

_molecules, 2023, doi:10.3390/molecules29010123_

Round 1
Reviewer 1 Report
Comments and Suggestions for Authors
In the present manuscript, the authors have prepared a series of bitopic ligands based on Fallypride with the aim of developing an S3R-selective compounds. Their activity and selectivity were evaluated in in vitro binding assays, while computational studies revealed the probable interaction network, leading to the above results.
The authors present some selective D3R ligands, which are shown in Figure 2, but it is not clear how these ligands are relevant to the current study, as the study focuses exclusively on compounds developed from Fallypride.
In Scheme 2, the preparation of the diastereomers and enantiomers of (R,S)-trans-2d and (R,S)-trans-2i should be shown. However, Scheme 2 is identical to Scheme 1, which shows the preparation of Fallypride-based ligands.
Lines 129-130: “As shown in Table 1 and 2, (R,S)-trans-2a has high potency in the 129 β-arrestin assay.” (R,S)-trans-2a is not listed in Table 2.
In the paragraph “Molecular dynamic simulation (MDS) studies” I miss the description of the binding site and the crucial interactions. Also, Figure 3 should be improved as the visibility is very poor, especially the bonds.
Lines 204-209: “(R,S)-trans-2a has a higher potency (IC50 = 0.8 ± 0.5 nM) to compete with dopamine than Fallypride ((IC50 = 1.7 ± 0.8 nM)) in the b-arrestin recruitment assay, which was unexpected given its lower affinity for the D3R relative to Fallypride. The reason for this discrepancy between receptor affinity and potency in the b-arrestin competition assay are not clear.” Has anything similar been reported in the literature for other ligands that might explain this observation?
How were the settings for the molecular docking studies validated?
It is not described how the purity of the synthesized compounds was assessed.
Author Response
Reviewer 1
In the present manuscript, the authors have prepared a series of bitopic ligands based on Fallypride with the aim of developing an D3R-selective compounds. Their activity and selectivity were evaluated in in vitro binding assays, while computational studies revealed the probable interaction network, leading to the above results.
- The authors present some selective D3R ligands, which are shown in Figure 2, but it is not clear how these ligands are relevant to the current study, as the study focuses exclusively on compounds developed from Fallypride.
Response: Figure 2 shows the different scaffolds that display high affinity and selectivity for the D3 versus D2 receptors. These structures were included to demonstrate that it is possible to generate D3-selective ligands.
- In Scheme 2, the preparation of the diastereomers and enantiomers of (R,S)-trans-2d and (R,S)-trans-2i should be shown. However, Scheme 2 is identical to Scheme 1, which shows the preparation of Fallypride-based ligands.
Response: We have corrected the Scheme 2 in the manuscript.
- Lines 129-130: “As shown in Table 1 and 2, (R,S)-trans-2a has high potency in the 129 β-arrestin assay.” (R,S)-trans-2a is not listed in Table 2.
Response: The data of (R,S)-trans-2a is in Table 1, we have corrected the sentence to “As shown in Table 1”.
- In the paragraph “Molecular dynamic simulation (MDS) studies” I miss the description of the binding site and the crucial interactions. Also, Figure 3 should be improved as the visibility is very poor, especially the bonds.
Response: We have added more description of the binding site and the key interactions in the section of MDS studies and improved the image quality and visibility of Figure 3.
- Lines 204-209: “(R,S)-trans-2a has a higher potency (IC50 = 0.8 ± 0.5 nM) to compete with dopamine than Fallypride ((IC50 = 1.7 ± 0.8 nM)) in the b-arrestin recruitment assay, which was unexpected given its lower affinity for the D3R relative to Fallypride. The reason for this discrepancy between receptor affinity and potency in the b-arrestin competition assay are not clear.” Has anything similar been reported in the literature for other ligands that might explain this observation?
Response: We are not aware of another compound displaying this interesting property. However, we must point out that most publications reporting new D3-selective compounds do not run this assay in the DA-competition mode. They only report results performing this assay in the agonist-binding mode to confirm that their compounds are antagonists at the D3 receptor.
- How were the settings for the molecular docking studies validated?
Response: The molecular docking studies were aimed to select a reasonable ligand binding pose in the binding pocket that reproduced the crystallographic ligand binding pose in the crystal structure, and then used it to set the initial step of the molecular dynamic simulations.
- It is not described how the purity of the synthesized compounds was assessed.
Response: The purity of all the synthesized compounds was measured by LC-MS. The data of LC-MS and spectrum are included in the supporting information.
Reviewer 2 Report
Comments and Suggestions for Authors
The Authors presented comprehensive studies on bitopic D3R ligands. The manuscript is well written and study has been planned in high quality scientific manner, thus it should be published in Molecules. However after consideration of some major and minor suggestions:
Major comments:
11) In the Introduction the Authors describe the two dopamine receptors classes. As the study concerns D2-like receptors (D2R and D3R), I would like to ask what about the D4R?There is only mentioned the this receptors exists. In my opinion it should be added explanation why the Authors do not consider this target as off-target in the selectivity considerations (may be pharmacological role)? The similarity of structures between D2R, D3R and D4R is very high. I have impression that more detailed information about D4 is missing especially that recently this receptors returned under interest of scientific community. The Authors do not present D4R binding results nor molecular modelling studies including this receptor. The comment from Authors would be useful.
22) It is not enough clear If Authors search for agonists or/and antagonists? Which would be useful from pharmacological point of view? And structurally is it possible to compare described novel bitopic ligands their similarity to known structure of antagonist/agonist? Also the introduction section do not bring information If exemplary application of dopamine ligands need agonists or antagonists.
33) The yields of reaction should be added to description of synthetic scheme
44) I suggest to check carefully experimental part as some important information is missing e.g. line 235 – in which temperature the reaction mixture was left overnight? What wat the form of compound 2 (line 240)?
55) As the study concern GPCR’s it would be useful to use in molecular modelling part e.g in brackets after the present number, the Ballesteros-Weinstein’s amino acid numbering as the huge amount of GPCR scientists used it
Minor comments:
11) The main text needs to be rechecked in terms of missing superscripts in D2R and D3R
22) R,S in stereochemistry usually is written in Italic
33) Line 154, 161 – should be 'pi-stacking' instead of 'pi-staking'
Author Response
Reviewer 2
The Authors presented comprehensive studies on bitopic D3R ligands. The manuscript is well written and study has been planned in high quality scientific manner, thus it should be published in Molecules. However after consideration of some major and minor suggestions:
Major comments:
- In the Introduction the Authors describe the two dopamine receptors classes. As the study concerns D2-like receptors (D2R and D3R), I would like to ask what about the D4R? There is only mentioned the this receptors exists. In my opinion it should be added explanation why the Authors do not consider this target as off-target in the selectivity considerations (may be pharmacological role)? The similarity of structures between D2R, D3R and D4R is very high. I have impression that more detailed information about D4 is missing especially that recently this receptors returned under interest of scientific community. The Authors do not present D4R binding results nor molecular modelling studies including this receptor. The comment from Authors would be useful.
Response: We typically do not measure the binding affinity of our compounds for the D4 receptor since this receptor has a low sequence homology to the D2 (41%) and D3 (41%) receptors. Consequently, benzamide analogs such as raclopride and remoxipride typically have a low affinity for the D4 receptor. We expect the same for our compounds.
- It is not enough clear If Authors search for agonists or/and antagonists? Which would be useful from pharmacological point of view? And structurally is it possible to compare described novel bitopic ligands their similarity to known structure of antagonist/agonist? Also the introduction section do not bring information if exemplary application of dopamine ligands need agonists or antagonists.
Response: We have added a sentence in the Introduction stating that we are interested in identifying D3R-selective antagonists for PET imaging studies.
- The yields of reaction should be added to description of synthetic scheme
Response: We have added the yield of the reaction to the synthetic schemes.
- I suggest to check carefully experimental part as some important information is missing e.g. line 235 – in which temperature the reaction mixture was left overnight? What is the form of compound 2 (line 240)?
Response: We have added the experimental temperature, room temperature, in line 235 in the experimental section. We have added a description of the physical forms of compound 2 (which is a series of compounds) in the experimental section of each compound.
- As the study concern GPCR’s it would be useful to use in molecular modelling part e.g in brackets after the present number, the Ballesteros-Weinstein’s amino acid numbering as the huge amount of GPCR scientists used it
Response: We have added the Ballesteros-Weinstein amino acid number as the superscript format to the amino acid residues that have mentioned in the main text and figures.
Minor comments:
- The main text needs to be rechecked in terms of missing superscripts in D2R and D3R
Response: We have checked the subscript of D2R and D3R and made the abbreviation consistent.
- R,S in stereochemistry usually is written in Italic
Response: We have corrected the written of stereochemistry.
- Line 154, 161 – should be 'pi-stacking' instead of 'pi-staking'
Response: We have corrected the misspelling.
Reviewer 3 Report
Comments and Suggestions for Authors
The current study aims to discovering D3R selective ligands and finally several such moderately selective ligands were yielded. The manuscript can be accepted after minor revisions.
1. The importance of finding a D3R selective ligand is not clearly described. The reviewer believes that the focus should be on unmet need on diseases that is highly related to D3R. Please modify the introduction section.
2. It seems that Scheme 1 and 2 are the same. Please clarify.
3. Please clearly show how these compounds, enantiomers and diastereomers of (R,S)-trans-2d and (R,S)-trans-2i, were prepared in the Scheme and also in the main text.
4. The modeling study is not adequate. Only the differences of alkane chain length were compared. At least two other compounds from the rest should be added and compared.
Author Response
Reviewer 3
The current study aims to discovering D3R selective ligands and finally several such moderately selective ligands were yielded. The manuscript can be accepted after minor revisions.
- The importance of finding a D3R selective ligand is not clearly described. The reviewer believes that the focus should be on unmet need on diseases that is highly related to D3R. Please modify the introduction section.
Response: Paragraph 2 describes the dysregulation of D2 and D3 receptors in different CNS disorders. Paragraph 3 describes the absence of D3-selective ligands and the need to have a D3R-selective radiotracer for studying this receptor independently of the D2R.
- It seems that Scheme 1 and 2 are the same. Please clarify.
Response: We have corrected the Scheme 2 in the manuscript.
- Please clearly show how these compounds, enantiomers and diastereomers of (R,S)-trans-2d and (R,S)-trans-2i, were prepared in the Scheme and also in the main text.
Response: We have added details in the results. “(R,S)-trans-1, (S,S)-trans-1 or (R,R)-trans-1, the diastereomers and enantiomers of (S,R)-trans-1, reacted with CDI in acetonitrile. The generated intermediate was further treated with the corresponding amines in toluene at 100 °C to give the diastereomers and enantiomers of (R,S)-trans-2d and (R,S)-trans-2i.”
We also added “the procedure for the synthesis of compounds 1 can be found in our previous report” and cited the reference in the experimental section.
- The modeling study is not adequate. Only the differences of alkane chain length were compared. At least two other compounds from the rest should be added and compared.
Response: We have included 2 more compounds in the modeling studies, and the results were updated in the Table 3 and Figure 4, as well as in the main texts for the section of molecular dynamic simulation studies.
Round 2
Reviewer 2 Report
Comments and Suggestions for Authors
Thank you very much for all the Author's responses. I would like to jus add the comment that it would be worth to take D4R under consideration in further studies (as off target), as known D2R ligands (e.g. haloperidol, thioridazine, chlorpromazine) have very high affinity also to D4R. However, I recommend to publish this interesting and well written manuscript in present form.
Author Response
Thank you for your comment and we will certainly do so.